# Modeling Long-Tail Relations in the Operating Room
# via In-Context Multimodal Learning

Boqiang Xu [* 1 2]   Wei Zhang [* 3]   Ding Ma [4]   Jian Liang [2 5]   Zhenan Sun [2 5]   Zhen Lei [1 2 5]

## Abstract

Operating room (OR) scene graph generation (SGG) enables holistic modeling of OR domains by encoding interactions among medical staff, tools, and equipment as triplet-based structured scene graphs. Although existing OR SGG methods demonstrate satisfactory overall performance, they exhibit substantially lower accuracy on long-tail categories compared to head categories in OR data. We introduce SGG-ICL, a novel framework that represents the first attempt to address the long-tail problem in OR SGG by leveraging in-context learning (ICL). SGG-ICL first identifies long-tail samples via an Adaptive Router module and selectively applies ICL only to these samples. This selective routing strategy enhances performance on long-tail categories without degrading head-category accuracy. Subsequently, SGG-ICL constructs a candidate pool through multimodal retrieval and then employs a trained MLLM Reranker to re-rank the candidates, selecting the most similar examples to the test sample for ICL. The reranker is supervised by IoU scores derived from annotated SGG triplets and exploits rich multimodal information to estimate pairwise sample similarity. Experimental results show that SGG-ICL improves accuracy on long-tail categories by 6.9%, while also achieving a 2.6% improvement in overall accuracy.

## 1. Introduction

The operating room (OR) is a highly dynamic, complex, and high-stakes environment that demands precise coordination, rapid decision-making, and accurate scene understanding to ensure safe and successful surgical outcomes (Lalys & Jannin, 2014) (Lalys & Jannin, 2014; Maier-Hein et al., 2017). In view of the critical need for automatic modeling of the OR to advance surgical data science (Özsoy et al., 2024a), a growing body of research has been devoted to the analysis of individual components of surgical workflows, including surgical phase recognition (Czempiel et al., 2021), action classification (Nwoye et al., 2020; 2022), and instrument usage detection (Jin et al., 2018).

Recently, attention has shifted toward holistic modeling of the OR domain through scene graph generation (SGG), which enables structured representations that capture the complete surgical scene and its multifaceted interactions via triplet-based encodings. For instance, the triplet *[head surgeon, patient, sawing]* is used to encode the interaction in which a head surgeon performs a sawing procedure on a patient. To advance the study of SGG, (Özsoy et al., 2025) introduced MM2SG, the first multimodal large language model (MLLM) designed for OR SGG. MM2SG incorporates a multimodal encoding module that transforms diverse inputs—including RGB images, segmentation masks, point cloud data, audio cues, speech, robotic system logs, and tracking information—into token embeddings. These embeddings are then jointly fine-tuned with a large language model (LLM) in an end-to-end manner to perform semantic scene graph generation.

Although MM2SG achieves promising performance, it still encounters distinct challenges. Specifically, OR data often exhibit a long-tailed distribution, with a few categories occurring frequently and a large number of categories appearing rarely or only once. Recent studies indicate that infrequently occurring knowledge in the training corpus can hinder LLMs' understanding of such knowledge (Wu et al., 2024b; Kandpal et al., 2023). As shown in Tab. 1, MM2SG attains an average F1 score of 0.664 on head classes (e.g., Close To, LyingOn), but only 0.289 on long-tail classes (e.g., Calibrating, Drilling), representing a substantial degradation of 0.375 for long-tail categories. This severe long-tail prob-

*Equal contribution  [1]the Centre for Artificial Intelligence and Robotics, Hong Kong Institute of Science & Innovation, Chinese Academy of Sciences, Hong Kong, China [2]the State Key Laboratory of Multimodal Artificial Intelligence Systems (MAIS), Institute of Automation, Chinese Academy of Sciences (CASIA), Beijing, China [3]School of Artificial Intelligence and Robotics, Hunan University, Changsha, Hunan Province, China [4]Graduate School of Informatics, Nagoya University, Nagoya, Aichi, Japan [5]the School of Artificial Intelligence, University of Chinese Academy of Sciences (UCAS), Beijing, China. Correspondence to: Zhen Lei <zhen.lei@ia.ac.cn>, Jian Liang <liangjian92@gmail.com>.

*Proceedings of the 43$^{rd}$ International Conference on Machine Learning*, Seoul, South Korea. PMLR 306, 2026. Copyright 2026 by the author(s).

*Table 1.* Scene graph generation results of MM2SG (Özsoy et al., 2025) on the MM-OR (Özsoy et al., 2025) dataset. Head classes are shown in the upper portion and long-tail classes in the lower portion, clearly illustrating the severe long-tail problem, as performance on long-tail classes drops substantially compared to head classes.

| Predicate | Count | Precision | Recall | F1 |
|---|---|---|---|---|
| CloseTo | 67148 | 0.803 | 0.638 | 0.711 |
| LyingOn | 45924 | 0.856 | 0.742 | 0.795 |
| Holding | 23487 | 0.791 | 0.414 | 0.543 |
| Manipulating | 14273 | 0.761 | 0.692 | 0.724 |
| Touching | 13963 | 0.614 | 0.633 | 0.623 |
| Preparing | 11681 | 0.700 | 0.845 | 0.764 |
| Assisting | 4635 | 0.426 | 0.270 | 0.331 |
| Sawing | 2383 | 0.918 | 0.741 | 0.820 |
| Calibrating | 1721 | 1.000 | 0.220 | 0.361 |
| Drilling | 1539 | 0.868 | 0.335 | 0.483 |
| Hammering | 269 | 0.588 | 0.323 | 0.417 |
| Suturing | 132 | 0.000 | 0.000 | 0.000 |
| Cutting | 125 | 0.000 | 0.000 | 0.000 |
| Cleaning | 113 | 0.286 | 0.667 | 0.400 |
| Scanning | 69 | 0.400 | 0.333 | 0.364 |
| **Head Classes** | - | 0.734 | 0.622 | 0.664 |
| **Long-Tail Classes** | - | 0.449 | 0.268 | 0.289 |

lem limits the applicability of existing OR SGG methods in real-world OR environments.

In this work, we present the first study to alleviate the long-tail problem in OR SGG. In-context learning (ICL) is a few-shot learning method in which LLMs are prompted by concatenating a small set of relevant examples with the test query, without any parameter updates (Brown et al., 2020). Kandpal *et al.* (Kandpal et al., 2023) further demonstrated that incorporating retrieval augmentation into ICL can reduce models' reliance on relevant pre-trained knowledge, presenting a promising approach for capturing the long-tail. However, incorporating ICL into OR SGG poses two key challenges: 1) **Long-tail identification**: how to determine whether a given test sample belongs to a long-tail category and thus requires augmentation. 2) **Long-tail retrieval**: as each OR SGG sample consists of rich multimodal information, including RGB images, point clouds, and speech signals, among others, making it nontrivial to retrieve relevant examples across such heterogeneous modalities.

To tackle these challenges, in this paper, we propose a novel method called SGG-ICL, which integrates ICL into OR SGG for long-tail relation modeling. For **long-tail identification**, we observe that long-tail categories (e.g., drilling, hammering, and cutting) can be recognized from multimodal information. Motivated by this observation, we propose an Adaptive Router that leverages Qwen3 to adaptively

route each test sample based on whether it belongs to a long-tail category. ICL is applied only to samples identified as long-tail, while samples associated with head categories are processed via direct inference without ICL. This enables SGG-ICL to improve performance on long-tail categories while avoiding adverse impact on head categories.

For **long-tail retrieval**, we design a two-stage multimodal retrieval framework that proceeds from coarse to fine. In the first stage, we leverage BGE-VL-MLLM (Zhou et al., 2025) to perform efficient multimodal retrieval and construct a pre-selected candidate pool. In the second stage, we train an MLLM Reranker using intersection-over-union (IoU) scores computed from annotated SGG triplets as supervision. This reranker leverages the comprehensive multimodal information of OR SGG samples to estimate pairwise sample similarity. The learned reranker is then used to re-rank the candidate pool and select the $top - k$ most relevant samples as in-context examples for ICL. We conduct experiments on the MM-OR (Özsoy et al., 2025) dataset. The results indicate that SGG-ICL yields a $6.9\%$ improvement in accuracy on long-tail categories, and also achieves a $2.6\%$ gain in overall accuracy.

The contributions of our paper are summarized as follows:

- We present the first work to alleviate the long-tail problem in OR SGG.

- We propose a novel framework called SGG-ICL, which mainly consists of an Adaptive Router for long-tail identification and an MLLM Reranker for long-tail retrieval.

- Extensive experiments demonstrate that our method consistently outperforms the state-of-the-art methods, achieving $6.9\%$ improvement in accuracy on long-tail relations.

**Conflict of Interest Disclosure.** The author Boqiang Xu is employed by the Centre for Artificial Intelligence and Robotics, Hong Kong Institute of Science & Innovation, Chinese Academy of Sciences, Hong Kong, China, which leads the development of SSG-ICL.

## 2. Related Work

**Operation Room Scene Graph Generation.** Scene graph generation (SGG) models encode the Operating Room (OR) into graphs, representing entities and their relationships. (Özsoy et al., 2022) was the first to introduce semantic scene graphs for holistic and semantic understanding and modeling of the OR domain. In the scene graph, nodes represent various actors and objects, such as medical staff, patients, and medical equipment, while edges encode the relationships among them. Moreover, (Özsoy et al., 2022)

constructed the first publicly available surgical SGG dataset, namely 4D-OR, and proposed an end-to-end neural-network-based pipeline for SGG. (Özsoy et al., 2024b) presents OR-acle, a vision–language framework for OR SGG that incorporates multi-view and temporal information and leverages external knowledge during inference to improve generalization to unseen surgical scenarios. Its effectiveness is further enhanced by a dedicated data augmentation strategy that increases training diversity. (Özsoy et al., 2025) introduces MM-OR, a large-scale and realistic multimodal OR SGG dataset. The dataset captures comprehensive OR scenes with diverse modalities and provides rich annotations. In addition, (Özsoy et al., 2025) proposes MM2SG, a multimodal vision–language model for OR SGG that effectively exploits multimodal inputs.

However, despite their promising performance on OR SGG, these methods still face significant challenges arising from long-tail distributions in the OR data. To the best of our knowledge, this is the first study to explicitly address the long-tail problem in OR SGG.

**Long-Tail Learning for ICL.** ICL (Brown et al., 2020) prompts LLMs by concatenating relevant examples with the test query, without requiring any parameter updates. Kandpal *et al.* (Kandpal et al., 2023) were the first to investigate the impact of long-tailed pre-training data distributions on LLM memorization, and they showed that retrieval augmentation with ICL is a promising strategy for substantially reducing LLMs' reliance on pre-trained knowledge. Recently, many methods have been dedicated to improving the effectiveness of retrievers in order to enhance the quality of ICL. For example, some approaches focus on fine-tuning retrievers for specific target domains, including UDR (Li et al., 2023), and LLM-R (Wang et al., 2024). Other works leverage GPT-based models to retrieve and re-rank samples through carefully designed prompts, such as Rerank (Sun et al., 2023) and SuRe (Kim et al., 2024). However, most existing methods are primarily designed for text-only scenarios and are not applicable to multimodal inputs such as images, point clouds, speech, and text. In addition, the distinctive triplet-based structure of semantic scene graphs prevents these methods from being directly applied to OR SGG.

To address these challenges, we propose a novel framework, termed SGG-ICL. Tailored to the unique characteristics of scene graphs, SGG-ICL incorporates a multimodal large language model reranker trained with IoU scores computed from annotated SGG triplets as supervision. This design facilitates more accurate and robust ICL for SGG under long-tailed data distributions.

## 3. Methodology

### 3.1. Overview

In this section, we introduce the structure of our SGG-ICL. First, we introduce an Adaptive Router that routes each test query to head or long-tail categories at inference time. Head-category queries are handled through direct inference, while long-tail queries are augmented with ICL. For long-tail queries, we first employ BGE-VL-MLLM (Zhou et al., 2025) to perform pre-selection and obtain a candidate pool. We then train an MLLM Reranker to compute similarity scores between the query and each candidate in the pre-selected pool, enabling re-ranking and selection of the $top-k$ most relevant candidates. Finally, the selected $top-k$ candidates are used to construct the ICL prompt for inference. The structure of the SGG-ICL is illustrated in Fig. 1.

### 3.2. Adaptive Router

As shown in Tab. 1, we define categories with fewer than 2,000 occurrences in the MM-OR dataset as long-tail categories, including Calibrating, Drilling, Hammering, Suturing, Cutting, Cleaning, and Scanning. Despite their low frequencies and the resulting poor performance of existing models, we observe that these long-tail categories correspond to specific surgical actions in the OR domain and can be recognized from multi-view RGB images. Therefore, we employ Qwen3-VL-30B as an Adaptive Router to determine whether a given query belongs to a long-tail category. Specifically, we implement the routing procedure by constructing a prompt template as:

> **Question:** You are given multi-view images from a robotic total or partial knee replacement surgery. {*Definition*} Please determine whether the images are currently performing the {*Action*}. Please answer "yes" or "no" and give the reason.
> **Answer:**

where {*Action*} denotes a long-tail surgical action, and {*Definition*} provides a detailed description of the corresponding action. The detailed definitions of the long-tail categories are provided in Tab. 2. For example, for the Cutting action, we adopt the following prompt:

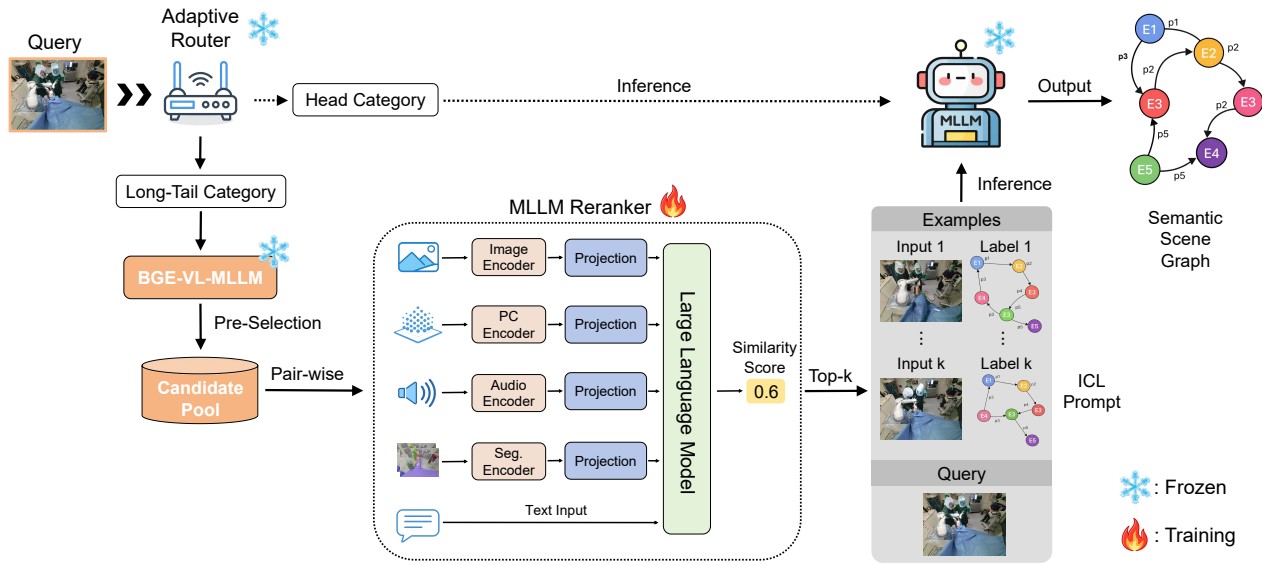

*Figure 1.* **Overall pipeline of SGG-ICL.** At inference time, an Adaptive Router first assigns each query to either a head or long-tail category. Head-category queries are processed via direct inference, while long-tail queries are augmented with ICL. For long-tail queries, BGE-VL-MLLM is used for candidate pre-selection, followed by an MLLM Reranker that computes query–candidate similarities to select the $top - k$ relevant examples, which are then used to construct the ICL prompt for inference.

**Question:** You are given multi-view images from a robotic total or partial knee replacement surgery. Cutting refers to the action in which a surgeon uses a scalpel to make an incision in the skin on the knee area of the patient. Please determine whether the images are currently performing the Cutting. Please answer "yes" or "no" and give the reason.
**Answer:**

Additionally, for Calibrating, we determine the action based on robot logs. Overall, we route test queries using six distinct prompts in conjunction with robot logs. If a query is identified as belonging to any long-tail category, it is forwarded to the subsequent ICL augmentation stage; otherwise, it is treated as a head-category query and routed to direct inference.

### 3.3. Pre-Selection

For test samples selected by the Adaptive Router as belonging to long-tail categories, we construct a candidate pool through multimodal retrieval using BGE-VL-MLLM (Zhou et al., 2025). Specifically, we first employ Qwen3-VL-30B to generate semantic textual descriptions for both the training samples and the selected test samples. Then, based on textual descriptions and multi-view images, BGE-VL-MLLM (Zhou et al., 2025) performs multimodal retrieval to identify the top-k training samples that are most similar to each selected test sample. These retrieved samples constitute the candidate pool.

*Table 2.* Detailed Definitions of Long-Tail Categories for the Adaptive Router.

| Action | Definition |
|---|---|
| Cleaning | Cleaning refers to the action in which medical staff use tweezers to hold a pad and wipe the surface of the patient. |
| Cutting | Cutting refers to the action in which a surgeon uses a scalpel to make an incision in the skin on the knee area of the patient. |
| Drilling | Drilling refers to the process in which the surgeon operates a powered drill to penetrate or shape the patient's bone or skin. |
| Scanning | Scanning refers to the process of using an imaging device (such as a C-arm or optical scanner) to capture precise intraoperative images of the patient's anatomy. |
| Suturing | Suturing refers to the process in which the surgeon closes the surgical incision. |
| Hammering | Hammering refers to the process in which the surgeon uses a surgical hammer or mallet to apply controlled force to an instrument or a patient. |

## 3.4. MLLM Reranker

Through the pre-selection stage, we obtain a candidate pool. However, BGE-VL-MLLM only supports multimodal retrieval based on text and images, and thus cannot fully exploit other modalities available in the OR domain, such as point clouds and audio signals. Moreover, BGE-VL-MLLM lacks an explicit understanding of scene graph structures, which limits its ability to accurately retrieve training samples that share similar scene graph structures with the test samples. To address these limitations, we propose an MLLM Reranker that can ingest rich multimodal inputs, including images, text, point clouds, and audio. By training the reranker with IoU scores computed from annotated SGG triplets as supervision, it is able to directly predict the scene graph similarity between two samples.

**The Architecture of MLLM Reranker.** Following (Özsoy et al., 2025), we adopt modality-specific encoders to effectively capture information from each modality:

- **Image Encoder:** We jointly process multi-view RGB-D images, high-resolution detail views, and robot interface recordings. Each image is encoded by a CLIP (Radford et al., 2021) vision encoder, and the resulting patch features are aggregated by a transformer-based image pooler to produce a fixed-length image representation. The image encoder and pooler are trained end-to-end with the LLM.

- **Point Cloud Encoder:** A Point Transformer V3 (Wu et al., 2024a) is used to encode 3D point clouds into a single token, trained end-to-end with the model.

- **Audio Encoder:** Audio signals are encoded using a pre-trained, frozen CLAP audio encoder (Elizalde et al., 2023), producing a single token per one-second audio segment.

- **Segmentation Encoder:** Segmentation masks are encoded with a lightweight CNN trained end-to-end, producing a single token embedding for each mask.

- **Text Input:** We preprocess the audio recordings and extract speech transcripts using a pre-trained speech-to-text model (Radford et al., 2023). The resulting transcripts, together with real-time robot system logs and tracking data, are concatenated as textual inputs and directly fed into the LLM.

All modality-specific token embeddings are projected into the language model embedding space via linear projection layers, concatenated, and fed into the LLM as a unified input sequence.

**Optimization of the MLLM Reranker.** Our goal is to enable the MLLM Reranker to predict the similarity between the scene graphs of two samples, thereby allowing effective re-ranking of the candidate pool. To this end, we first define a similarity measure between scene graphs and use it as the training objective for the MLLM Reranker. Since our primary objective is to improve SGG accuracy for long-tail categories, the similarity definition is designed to satisfy the following criteria:

1. Two scene graphs that contain the same long-tail triplets should receive substantially higher similarity scores.

2. Two scene graphs that share a larger number of identical triplets should also be assigned higher similarity scores.

Formally, let a scene graph $\mathcal{G}$ be represented as a set of triplets $\mathcal{G} = \{(s, r, o)\}$, where $s$, $r$, and $o$ denote the subject, relation, and object, respectively. Given two scene graphs $\mathcal{G}_1$ and $\mathcal{G}_2$, we define their similarity using a weighted intersection-over-union (IoU) over triplets:

$$Sim(\mathcal{G}_1, \mathcal{G}_2) = \frac{\sum\limits_{(s,r,o)\in\mathcal{G}_1\cap\mathcal{G}_2} w(r)}{\sum\limits_{(s,r,o)\in\mathcal{G}_1\cup\mathcal{G}_2} w(r)} \qquad (1)$$

where $w(r)$ is a relation-specific weighting factor associated with the relation $r$, and two triplets $(s, r, o)$ are considered identical if their subject, relation, and object exactly match. To emphasize long-tail relations, we define the weight $w(r)$ as the inverse frequency of the corresponding relation in the training set:

$$w(r) = \frac{1}{f(r)}, \quad f(r) = \frac{N_r}{N}, \qquad (2)$$

where $N_r$ denotes the number of scene graph triplets in the training set that contain relation $r$, as summarized in Tab. 1, and $N$ denotes the total number of scene graph triplets in the training set. Through this design, Eq. (2) allocates higher weights to long-tail relations. Eq.(1) assigns higher similarity scores to pairs of scene graphs that share identical triplets, with additional emphasis on triplets containing long-tail relations.

We sample instances from the annotated SGG training set and construct the training data for the MLLM Reranker by forming sample pairs in a pairwise manner. The ground-truth label for each pair is defined as

$$\tilde{Sim}(G_1, G_2) = \mathrm{round}(Sim(G_1, G_2), 1), \qquad (3)$$

where $\mathrm{round}(\cdot, 1)$ denotes rounding to one decimal place. During sampling, we ensure a balanced ratio of 1:1 between

long-tail and head-category samples, and we additionally strive to collect pairs covering a diverse range of similarity scores to improve model generalization.

The final training loss of the MLLM Reranker is defined as:

$$\mathcal{L} = \frac{1}{|\mathcal{P}|} \sum_{(G_1, G_2) \in \mathcal{P}} \left\| f_\theta(G_1, G_2) - \tilde{Sim}(G_1, G_2) \right\|_2^2,$$

(4)

where $\mathcal{P}$ denotes the set of scene graph pairs used for training, and $f_\theta(G_1, G_2)$ is the similarity score predicted by the MLLM Reranker.

The prompt used for the MLLM Reranker is defined as follows:

---

**Question:** Sample A: {*Sample A*}, Sample B: {*Sample B*}. On a 0–1 scale, please score the similarity between the scene graphs of Sample A and Sample B. A higher score indicates greater similarity. Consider object nodes, relations, and the overall topology.
**Answer:**

---

where {*Sample A*} and {*Sample B*} denote the multimodal information of the two samples, including multi-view RGB-D images, point clouds, audio signals, and other relevant modalities. After computing similarity scores between each test sample and its candidate pool using the MLLM Reranker, we compute the average candidates similarity score for each test sample. A margin threshold $\Gamma$ is then applied for filtering: test samples whose average candidate similarity score falls below $\Gamma$ are excluded from subsequent ICL inference. This is to ensure the high quality of the constructed ICL prompt.

### 3.5. In-Context Learning Prompt Design

Through the MLLM Reranker, we identify the $top - k$ training samples whose scene graphs are most similar to the test query. These $top - k$ samples are then used to construct the ICL prompt as follows:

---

**Example 1:** Question: {$Example_1$}, Answer: {$Label_1$}.
   **...**
**Example k:** Question: {$Example_k$}, Answer: {$Label_k$}.
**Question:** {$Query$}
**Answer:**

---

where {$Example_i$}$_{i=1}^k$ denote the multimodal inputs of the $top - k$ samples, and {$Label_i$}$_{i=1}^k$ correspond to their associated scene graph annotations. {$Query$} represents the multimodal input of the test query. Finally, we input the constructed ICL prompt to the SGG model for inference.

## 4. Experiments

### 4.1. Experimental Setup

**Datasets.** Currently, the datasets available for OR SGG include 4D-OR (Özsoy et al., 2022) and MM-OR (Özsoy et al., 2025). Among them, 4D-OR is an earlier and relatively small-scale dataset, and its long-tail distribution is not obvious. Therefore, we conduct our experiments on the MM-OR dataset. MM-OR is currently the largest multimodal OR SGG dataset, including multi-view RGB-D data, high-resolution RGB views, a low-exposure tracking camera, audio, speech transcripts, real-time robotic system logs, screen recordings of the robot interface, and infrared tracking. It captures 17 full-length (approximately 90 minutes) and 22 short clips of robotic total and partial knee replacement surgeries. Overall, the dataset contains 92,983 total time points and 25,277 annotations, amounting to a total of 500 GB of data.

**Implementation Details.** In our experiments, we used four A100 GPUs with 80 GB memory each. We initialize our MLLM Reranker with pretrained LLaVA-7B (Liu et al., 2023) weights, utilizing Vicuna-7B (Chiang et al., 2023) as the LLM, and train it for 50 epochs with a batch size of 12. For efficient fine-tuning, we apply LoRA (Hu et al., 2022) to the MLLM Reranker and fine-tune the last 12 layers of the image encoder to adapt to the OR domain, with the number of image tokens set to $N = 576$. The margin threshold $\Gamma$ is set to $0.4$. We use BGE-VL-MLLM (Zhou et al., 2025) to retrieve the $top - 20$ samples to form the candidate pool, and then employ the MLLM Reranker to select the $top - 1$ sample as the ICL example.

### 4.2. Main Results

We report our results on MM-OR (Özsoy et al., 2025) dataset in Table 3. We present the accuracy for each category, where the upper part of the table corresponds to head classes and the lower part corresponds to long-tail classes. We also report the accuracy on head classes, the accuracy on long-tail classes, and the average accuracy. We compare our method with various representative baseline models, including PSG (Yang et al., 2022), Oracle (Özsoy et al., 2024b), and MM2SG (Özsoy et al., 2025). The results show that our method yields consistent improvements across all long-tail classes, with the average F1 score for long-tail classes increasing by at least $0.069$ and the overall average F1 score across all classes improving by at least $0.026$, while the F1 score for head classes decreases slightly by $0.006$. This decrease in head class accuracy is due to SGG-ICL misclassifying a small number of head class examples as long-tail classes and selecting incorrect examples to construct prompts during the ICL process, which negatively impacted the accuracy of the head classes. The experimental results demonstrate that our method significantly improves the long-

*Table 3.* Scene graph generation results on MM-OR (Özsoy et al., 2025) dataset. Head classes are shown in the upper portion and long-tail classes in the lower portion. SGG-ICL shows comparable performance on head classes, with significant improvements in long-tail classes and overall average scores.

| | PSG | Oracle | MM2SG | | | SGG-ICL (Ours) | | |
|---|---|---|---|---|---|---|---|---|
| Predict | F1 | F1 | Precision | Recall | F1 | Precision | Recall | F1 |
| Assisting | - | - | 0.426 | 0.270 | 0.331 | 0.402 | 0.277 | 0.328 |
| CloseTo | - | - | 0.803 | 0.638 | 0.711 | 0.798 | 0.631 | 0.705 |
| Holding | - | - | 0.791 | 0.414 | 0.543 | 0.784 | 0.395 | 0.525 |
| LyingOn | - | - | 0.856 | 0.742 | 0.795 | 0.864 | 0.754 | 0.805 |
| Manipulating | - | - | 0.761 | 0.692 | 0.724 | 0.760 | 0.670 | 0.712 |
| Preparing | - | - | 0.700 | 0.845 | 0.764 | 0.682 | 0.846 | 0.755 |
| Sawing | - | - | 0.918 | 0.741 | 0.820 | 0.902 | 0.736 | 0.811 |
| Touching | - | - | 0.614 | 0.633 | 0.623 | 0.620 | 0.615 | 0.617 |
| None | - | - | 0.964 | 0.989 | 0.976 | 0.963 | 0.989 | 0.976 |
| Calibrating | - | - | 1.000 | 0.220 | 0.361 | 0.937 | 0.245 | 0.388 |
| Cleaning | - | - | 0.286 | 0.667 | 0.400 | 0.310 | 0.690 | 0.428 |
| Cutting | - | - | 0.000 | 0.000 | 0.000 | 0.000 | 0.000 | 0.000 |
| Drilling | - | - | 0.868 | 0.335 | 0.483 | 0.752 | 0.365 | 0.491 |
| Hammering | - | - | 0.588 | 0.323 | 0.417 | 0.650 | 0.562 | 0.603 |
| Scanning | - | - | 0.400 | 0.333 | 0.364 | 0.420 | 0.362 | 0.389 |
| Suturing | - | - | 0.000 | 0.000 | 0.000 | 0.642 | 0.125 | 0.209 |
| **Head Classes** | - | - | 0.759 | 0.663 | 0.699 | 0.753 | 0.657 | 0.693 |
| **Long-Tail Classes** | - | - | 0.449 | 0.268 | 0.289 | 0.530 | 0.336 | 0.358 |
| **Macro Average** | 0.328 | 0.472 | 0.623 | 0.490 | 0.520 | 0.655 | 0.516 | 0.546 |

*Table 4.* The effectiveness of the Adaptive Router.

| | Head Classes | | | Long-Tail Classes | | | Macro Average | | |
|---|---|---|---|---|---|---|---|---|---|
| | Precision | Recall | F1 | Precision | Recall | F1 | Precision | Recall | F1 |
| w/o Adaptive Router | 0.678 | 0.587 | 0.622 | 0.329 | 0.177 | 0.202 | 0.504 | 0.382 | 0.412 |
| SGG-ICL | **0.753** | **0.657** | **0.693** | **0.530** | **0.336** | **0.358** | **0.655** | **0.516** | **0.546** |

tail accuracy in OR SGG, with only a slight impact on the head classes.

### 4.3. Analysis

**The effectiveness of Adaptive Router.** In Table 4, we evaluate the effectiveness of the Adaptive Router on the MM-OR dataset. From the results, we observe that removing the Adaptive Router leads to a drop of 0.071 in the F1 score for head classes, a decrease of 0.156 in the F1 score for long-tail classes, and a reduction of 0.134 in the macro-averaged F1 score. These findings show that removing the Adaptive Router and applying ICL to all samples leads to degraded performance. This is because head classes contain many relations, such as 'CloseTo', 'LyingOn', and 'Holding', which frequently occur in the OR domain and make it difficult to select appropriate in-context examples. As a result, indiscriminately applying ICL to all samples can

instead reduce the accuracy of OR SGG.

Furthermore, we present in Fig. 2 the confusion matrix of classification on the test set of the MM-OR dataset. The horizontal axis denotes the predicted labels, while the vertical axis represents the ground-truth labels. From the experimental results, we observe that: 1) The Adaptive Router misclassifies some test samples from head classes as long-tail classes, which subsequently are routed to ICL inference. This misclassification introduces a certain impact on the performance of SGG-ICL for head-class prediction. 2) The Adaptive Router is capable of effectively recalling a portion of long-tail classes, which in turn improves the performance of SGG-ICL on long-tail class predictions. Nevertheless, the recall is still relatively limited; in future work, we aim to further enhance the recall of long-tail classes to achieve improved model performance.

**The effectiveness of MLLM Reranker.** In Table 5, we

*Table 5.* The effectiveness of MLLM Reranker. Top-1 Accuracy refers to the accuracy with which the test sample and the selected top-1 sample share the same long-tail class relationship.

| | Head Classes | | | Long-Tail Classes | | | Macro Average | | | TOP-1 Accuracy (%) |
|---|---|---|---|---|---|---|---|---|---|---|
| | Precision | Recall | F1 | Precision | Recall | F1 | Precision | Recall | F1 | |
| w/o MLLM Reranker | 0.752 | 0.654 | 0.690 | 0.368 | 0.331 | 0.298 | 0.584 | 0.513 | 0.519 | 48.9 |
| SGG-ICL | **0.753** | **0.657** | **0.693** | **0.530** | **0.336** | **0.358** | **0.655** | **0.516** | **0.546** | **72.8** |

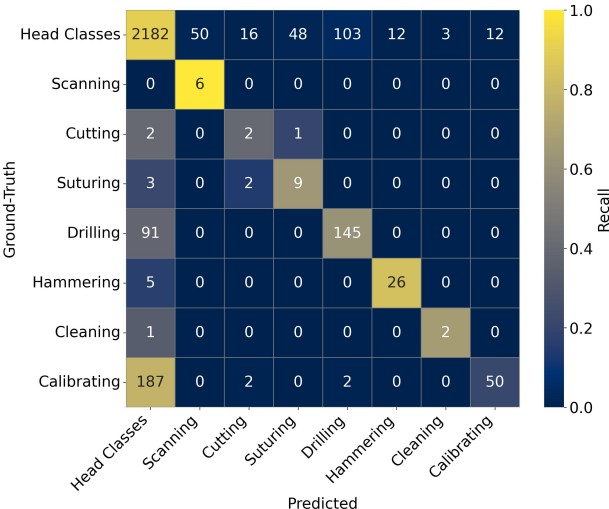

*Figure 2.* Confusion matrix of the Adaptive Router.

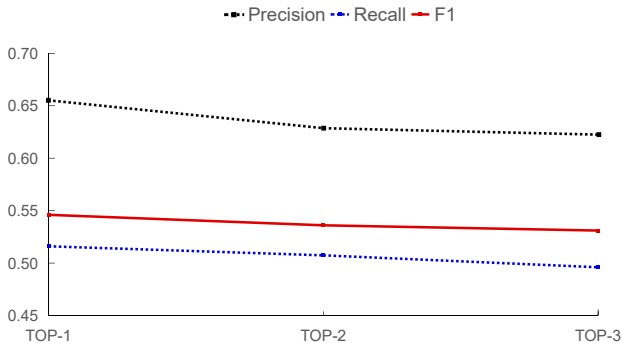

*Figure 3.* Analysis of the parameter $Top - k$.

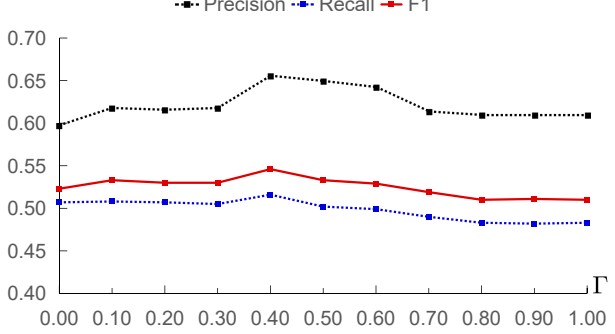

*Figure 4.* Analysis of the parameter $\Gamma$.

evaluate the effectiveness of the MLLM Reranker on the MM-OR dataset, where Top-1 Accuracy refers to the accuracy with which the test sample and the selected top-1 sample share the same long-tail class relationship. From the results, we observe that removing the MLLM reranker leads to a decrease of $0.06$ in the F1 score for long-tail classes, a reduction of $0.027$ in the macro-averaged F1 score, and a drop of $23.9\%$ in Top-1 Accuracy. These findings indicate that the MLLM Reranker substantially improves the Top-1 accuracy, thereby enhancing the SGG performance of SGG-ICL on long-tail class relationships.

**The effectiveness of parameter** $Top - k$**.** During the ICL inference stage, the $Top - k$ samples ranked by the MLLM Reranker are selected as ICL examples and used for inference. We evaluate the effects of the parameter $Top - k$ in Fig. 3. From the results, we observe a consistent degradation in model performance as $Top - k$ increases. Although a larger $Top - k$ provides more in-context learning examples, it also incorporates examples with lower similarity to the test sample, which in turn negatively affects the effectiveness of ICL inference.

**The impact of parameter** $\Gamma$**.** After obtaining the similarity scores between each test sample and the samples in the candidate pool using the MLLM Reranker, we compute the

average similarity score of the corresponding candidates for each test sample. A margin threshold $\Gamma$ is then applied for filtering; test samples whose average candidate similarity score falls below $\Gamma$ are excluded from subsequent ICL inference. We evaluate the effects of the parameter $\Gamma$ in Fig. 4. When $\Gamma$ is set to a low value, the filtering effect of the MLLM Reranker is weakened, and SGG-ICL performs ICL inference on most test samples that are classified as long-tail by the Adaptive Router. In contrast, when $\Gamma$ is set to a high value, the MLLM reranker allows ICL inference for only a small subset of test samples identified as long-tail by the Adaptive Router. From the results, we observe that the performance of SGG-ICL first improves and then degrades as $\Gamma$ increases, reaching its peak at $\Gamma = 0.40$.

**Inference Time.** The inference time of MM2SG is approximately 1 second per sample. For head-class samples, our

method takes approximately 3 seconds per sample, since they only go through the routing stage and no ICL inference is performed. For long-tail-class samples, which account for only about one-tenth of the dataset, our method takes approximately 5 seconds per sample.

## 5. Conclusion

In this paper, we propose SGG-ICL, the first approach to address the long-tail problem in OR SGG through ICL. SGG-ICL first performs long-tail identification via an Adaptive Router. This selective routing strategy enables SGG-ICL to apply ICL inference only to long-tail test samples, thereby enhancing performance on long-tail categories without degrading accuracy on head categories. Subsequently, SGG-ICL leverages multimodal retrieval and the MLLM Reranker to enable effective selection of in-context examples from heterogeneous OR data. The MLLM Reranker is supervised by IoU scores derived from annotated SGG triplets and exploits rich multimodal information to estimate pairwise sample similarity. Extensive experiments demonstrate that SGG-ICL effectively addresses the long-tail problem in OR SGG.

## Acknowledgements

This work was supported in part by the New Generation Artificial Intelligence-National Science and Technology Major Project (No. 2025ZD0123501), in part by the InnoHK Program, and in part by the National Natural Science Foundation of China under Grants U2441251, 62276256, 62306313, and U1836217.

## Impact Statement

This paper presents work whose goal is to advance the field of Machine Learning. There are many potential societal consequences of our work, none of which we feel must be specifically highlighted here.

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
