# OpenReview forum: "Modeling Long-Tail Relations in the Operating Room via In-Context Multimodal Learning"
_ICML.cc/2026/Conference — ICML 2026 regular_

### Official Review · Reviewer_xKTH · 2026-03-10

**Soundness:** 2
**Presentation:** 2
**Significance:** 2
**Originality:** 3
**Overall Recommendation:** 3
**Confidence:** 4

**Summary:**

This paper proposes SGG-ICL, a framework for improving long-tail relation prediction in operating room scene graph generation (OR-SGG). The approach introduces a selective in-context learning pipeline designed specifically for rare relations. The system first employs an Adaptive Router to identify whether an input sample belongs to long-tail categories. Only those samples predicted as long-tail are augmented with in-context examples. To obtain relevant examples, the method performs multimodal retrieval followed by a trained MLLM reranker that estimates scene graph similarity between samples using a weighted IoU over triplets. The selected examples are then used to construct prompts for in-context inference. Experiments on the MM-OR dataset demonstrate improved performance on long-tail relations while largely maintaining head-class accuracy.

**Compliance With Llm Reviewing Policy:**

Affirmed.

**Key Questions For Authors:**

It would be helpful to include a retrieval-only baseline where the label of the top retrieved example is directly used as the prediction.

**Limitations:**

no, this paper doesn't include a limitation section

**Strengths And Weaknesses:**

**Strength**

1.The paper addresses the long-tail issue in operating room scene graph generation, which is a realistic and impactful challenge in surgical data understanding.
2. Practical system design. The selective in-context learning strategy combined with an Adaptive Router is intuitive and avoids unnecessary prompting for frequent head classes.
3. Experiments and ablation studies demonstrate consistent improvements on long-tail relations and highlight the contributions of the reranker and routing components.

**Weakness**
1. The proposed framework essentially functions as a multi-stage cascade system similar to a mixture-of-experts or ensemble pipeline, consisting of an Adaptive Router, retrieval module, reranker, and final inference stage. When adapting the framework to new datasets or domains, a new Adaptive Router would likely need to be trained to identify long-tail categories specific to that dataset. This requirement increases the training complexity and reduces the ease of extending the system to new environments. The Adaptive Router also depends on a large vision-language model, which may introduce additional computational cost and system complexity during inference.
2. Limited applicability to zero-shot scenarios. The proposed framework relies on retrieving annotated samples that contain similar relations from the training set. Consequently, when new relations or previously unseen surgical actions appear in the data, the retrieval stage may fail to provide suitable demonstrations. This could limit the applicability of the approach in zero-shot or open-world settings where new relations emerge without labeled examples.
3. Limited evaluation across datasets. The experimental evaluation is conducted only on a single dataset (MM-OR). Although the authors mention that other operating room scene graph datasets are relatively small, the lack of additional benchmarks makes it difficult to assess the generalization ability of the proposed method. Evaluating the approach on additional datasets or related scene graph tasks would provide stronger evidence of its effectiveness and robustness.

---

> ### Author Rebuttal · Authors · 2026-03-29
>
> 1. Our Adaptive Router uses Qwen3-VL-30B, while the MLLM Reranker uses LLaVA-7B. The Adaptive Router is kept frozen, and only the MLLM Reranker is trained. Therefore, there is no need to worry that “a new Adaptive Router would likely need to be trained to identify long-tail categories specific to that dataset.” The inference time of MM2SG is approximately 1 second per sample. For head-class samples, our method takes approximately 3 seconds per sample, since they only go through the routing stage and no ICL inference is performed. For long-tail-class samples, which account for only about one-tenth of the dataset, our method takes approximately 5 seconds per sample.
>
> 2. Our method is intended to improve the performance of long-tail classes through ICL, rather than to address a zero-shot setting. In the OR SGG scenario, the actions, medical instruments, persons, and relations involved in surgical procedures are known in advance and have already appeared in previous videos of the same type of surgery. Therefore, this task usually does not involve a zero-shot setting.
>
> 3. Thank you for your suggestion. Our method is designed for the SGG task in operating-room scenes. At present, the only publicly available OR SGG datasets are MM-OR and 4D-OR. Among them, 4D-OR is an earlier and relatively small-scale dataset, and its long-tail distribution is not prominent. Therefore, we conduct our experiments on the MM-OR dataset.
>
>     A key characteristic of the SGG task in operating-room scenes is that long-tail relations in the scene graph can be identified using multimodal information, such as video and medical instrument information. This makes it possible to use the Adaptive Router to identify and route long-tail samples; however, this assumption may not hold in other scenarios. Moreover, like our method, other OR SGG works, such as Oracle and MM2SG, are also evaluated only on OR SGG datasets.
>
>     Overall, since our model is specifically designed for the SGG task in operating-room scenes, we believe that validating it on datasets from other scenarios would not be relevant to our goal.
>
> 4. We add a retrieval-only baseline. The experimental results are as follows:
> | Method | Head Classes |  |  | Long-Tail Classes |  |  | Macro Average |  |  |
> |:---:|:---:|:---:|:---:|:---:|:---:|:---:|:---:|:---:|:---:|
> |  | Precision | Recall | F1 | Precision | Recall | F1 | Precision | Recall | F1 |
> | Retrieval-only | 0.625 | 0.572 | 0.591 | 0.467 | 0.292 | 0.301 | 0.555 | 0.449| 0.464 |
> | SGG-ICL | **0.753** | **0.657** | **0.693** | **0.530** | **0.336** | **0.358** | **0.655** | **0.516** | **0.546** |
>
> A retrieval-only method would severely impair the accuracy of head relations in scene graph prediction. Moreover, for a scene-graph triplet [object, subject, relation], a prediction is considered correct only if all three elements are predicted correctly. Even if two scenes share the same relation, they may still involve different objects and subjects. Therefore, directly using the top retrieved example as the prediction is not a good solution.

---

> > ### Author Rebuttal · Reviewer_xKTH · 2026-04-07
> >
> > The rebuttal clarifies several implementation details and provides additional experimental evidence, particularly regarding the frozen Adaptive Router and the retrieval-only baseline. These help improve the clarity and empirical support of the work. However, my main concerns remain largely unchanged. The method appears to be strongly tailored to the operating-room SGG setting and relies on domain-specific assumptions, which limits its generalizability. In addition, the lack of evaluation beyond a single dataset and the absence of support for unseen relations further constrain its broader applicability. Therefore, my overall assessment and score remain unchanged.

---

> > > ### Author Response · Authors · 2026-04-07
> > >
> > > Thank you for the reviewer’s response. The objective of our paper is to address the long-tail issue in the operating room scenario through an in-context learning approach. Accordingly, our method is intentionally designed under operating room domain-specific assumptions. We would like to clarify that our goal is not to propose a universal solution for long-tail problems across all scenarios. For this reason, our experiments were conducted exclusively on operating room-related datasets. In addition, our experimental setup and datasets are consistent with prior studies in the operating room SGG [1,2,3,4]. For this reason, we kindly ask that the paper be assessed within the scope of the operating room SGG domain, rather than against broader settings beyond the target application. Thank you.
> > >
> > > [1] Özsoy, Ege, et al. "4d-or: Semantic scene graphs for or domain modeling." International conference on medical image computing and computer-assisted intervention. 2022.
> > >
> > > [2] Özsoy, Ege, et al. "Holistic or domain modeling: a semantic scene graph approach." International Journal of Computer Assisted Radiology and Surgery. 2024.
> > >
> > > [3] Özsoy, Ege, et al. "Oracle: Large vision-language models for knowledge-guided holistic or domain modeling." International Conference on Medical Image Computing and Computer-Assisted Intervention. 2024.
> > >
> > > [4] Özsoy, Ege, et al. "Mm-or: A large multimodal operating room dataset for semantic understanding of high-intensity surgical environments." Proceedings of the Computer Vision and Pattern Recognition Conference. 2025.

---

### Official Review · Reviewer_feV8 · 2026-03-12

**Soundness:** 2
**Presentation:** 3
**Significance:** 2
**Originality:** 2
**Overall Recommendation:** 3
**Confidence:** 3

**Summary:**

This paper proposes SGG-ICL, a framework that addresses the long-tail relation problem in Operating Room (OR) Scene Graph Generation (SGG) via in-context learning (ICL). The method consists of three components: (1) an Adaptive Router using Qwen3-VL to classify test samples as head or long-tail; (2) a coarse-to-fine multimodal retrieval pipeline (BGE-VL-MLLM + MLLM Reranker) to select the most relevant training examples; and (3) an ICL prompt constructed from the top-k retrieved examples for inference. The MLLM Reranker is trained with a weighted IoU-based similarity score as supervision. Experiments on MM-OR show +6.9% accuracy on long-tail classes and +2.6% overall.

**Compliance With Llm Reviewing Policy:**

Affirmed.

**Final Justification:**

Thanks for the rebuttal. I would maintain my score.

**Key Questions For Authors:**

The Adaptive Router runs six binary prompts per sample using Qwen3-VL-30B. What is the per-sample inference time compared to direct MM2SG inference? Is the system feasible for real-time or near-real-time OR use?

Several long-tail categories (Cutting, Suturing) still achieve F1 = 0.000 after applying SGG-ICL. What is the underlying reason — router failure, retrieval failure, or insufficient training examples? Would the method benefit from data augmentation for these extreme-tail categories?

PSG and Oracle results are missing for per-class metrics in Table 3. Why are these baselines not fully reported, and how does SGG-ICL compare to them on long-tail classes specifically?

**Limitations:**

Partially addressed. The paper acknowledges the slight drop in head-class accuracy and the limited recall of the Adaptive Router. However, the authors do not discuss failure cases for zero-F1 long-tail classes, or potential risks of misidentification in a safety-critical OR environment. These should be addressed more explicitly.

**Strengths And Weaknesses:**

Practical motivation and clear problem formulation. The long-tail challenge in OR SGG is well-motivated with concrete statistics (F1 gap of 0.375 between head and long-tail classes in MM2SG), and the selective routing design is sensible — applying ICL only to long-tail samples avoids degrading head-class performance.

Using weighted triplet IoU as a scene-graph-aware similarity signal for training the MLLM Reranker is a non-trivial and well-reasoned contribution, going beyond generic embedding-based retrieval.

Experiments are conducted on a single dataset (MM-OR), with comparisons only against OR-specific baselines (PSG, Oracle, MM2SG). PSG and Oracle results are not even reported per-class (shown as "–"), making the comparison incomplete. Generalizability to other multimodal or long-tail SGG settings is not demonstrated.

The pipeline invokes Qwen3-VL-30B six times per test sample for routing, plus BGE-VL-MLLM retrieval and MLLM Reranker scoring. No latency or computational cost analysis is provided, which is critical for a real OR deployment context.

---

> ### Author Rebuttal · Authors · 2026-03-29
>
> 1. The inference time of MM2SG is approximately 1 second per sample. For head-class samples, our method takes approximately 3 seconds per sample, since they only go through the routing stage and no ICL inference is performed. For long-tail-class samples, which account for only about one-tenth of the dataset, our method takes approximately 5 seconds per sample.
>
> 2. For some long-tail classes, such as Cutting, there are only four samples in the test set. Our adaptive router successfully routed two of them as long-tail classes and retrieved the correct ICL demonstrations for them. However, the subsequent ICL inference still failed to predict the Cutting relation contained in those samples. We think the failure is due to the fact that, for certain extremely scarce long-tail classes, the final SGG model is still unable to make correct inferences even with ICL.
>
> 3. The PSG and Oracle papers do not report per-class scores. MM2SG is currently the best-performing OR SGG model. As shown in Table 3, the Macro Average F1 score of MM2SG is 19.2% and 4.8% higher than those of PSG and Oracle, respectively. Therefore, we believe that improving the performance of long-tail classes on top of MM2SG is sufficient to demonstrate the effectiveness of our method.

---

### Official Review · Reviewer_BZEN · 2026-03-12

**Soundness:** 3
**Presentation:** 3
**Significance:** 3
**Originality:** 3
**Overall Recommendation:** 4
**Confidence:** 4

**Summary:**

The paper studies long-tail prediction for Operating Room (OR) scene graph generation, where rare surgical actions (long-tail categories) like calibrating, cleaning, cutting are harder to predict. The authors propose SSG-ICL which uses an Adaptive Router module (decides whether a sample is long-tail or not and selectively apply ICL to those samples) and Multi-modal retriever + MLLM reranker (estimates scene-graph similarity using a weighted IoU-style score over triplets, with extra emphasis on rare relation). Top-k examples are used for ICL.

Experiments on MM-OR dataset show improved long-tail F1 over MM2SG, with small degradation on head classes but modest gain in macro average performance. The primary contribution is selective ICL and multimodal reranking in OR scene graph generation for long-tail relation handling.

**Compliance With Llm Reviewing Policy:**

Affirmed.

**Final Justification:**

The authors have adequately addressed the reviewers’ comments and clarified several points in the revised manuscript. However, while these changes improve readability and presentation, they do not substantially alter the core contributions or overall impact of the work. Therefore, my evaluation and score remain unchanged.

**Key Questions For Authors:**

1.	Can you report **variance across multiple runs** or random seeds for the main MM-OR results? If the gains are stable, that would increase my confidence in the method.
2.	How does SGG-ICL compare against **simpler retrieval baselines**, such as random demonstrations, nearest-neighbor retrieval without the reranker, or retrieval based only on image/text similarity? This would clarify how much each component contributes.
3.	What is the computational cost at inference time compared with MM2SG alone? Since the method adds routing, retrieval, and reranking.
4.	Why do some rare classes, especially Cutting, remain at zero F1? Is this due to data scarcity, router failure, retrieval failure, or the final SGG model? A failure analysis would help in understanding the corner cases of model and its robustness.

**Limitations:**

No. The paper should discuss limitations more explicitly, especially dependence on a single dataset, instability risk in retrieval/ICL, inference cost from multiple large models, and the fact that some rare classes still remain essentially unsolved. A brief discussion of possible clinical misuse or safety-critical settings would also be helpful in surgical domain works.

**Strengths And Weaknesses:**

**Soundness:** The work is very technical, and the problem formulation has a clear motivation. The authors identify the limitations of long-tail distribution in OR scene graph generation.  The paper target a failure mode of prior OR SGG systems. The idea of routing only likely long-tail cases into ICL is sensible, because applying retrieval and prompting to every sample could easily hurt common relations. The overall framework: route, retrieve, rerank and prompt makes it a easy to understand procedure.
The proposed components are technical, with supporting results showing improvement on long-tail classes. The ablations on the router and reranker are useful and support the claim that both parts matter.

**Weakness:**
1.  The evidence is limited. The evaluation is on a **single dataset**, and there is no report of *statistical variance* across seeds or runs, which matters because retrieval-based and LLM-based pipelines can be unstable.
2. There are not enough simpler baselines to isolate whether the gains really come from the proposed design choices.
3. Compare against simpler alternatives such as random ICL examples, nearest-neighbor retrieval without reranking (Khandelwal et al., 2020: Generalization through Memorization: Nearest Neighbor Language Models), or a non-ICL long-tail balancing strategy (Class Balance Loss (Cui et al., 2019)/Label-Distribution-Aware Margin Loss (Cao et al., 2019)).
4. Missing justification on why some long-tail classes (Cutting) remain at or near zero performance, which weakens the claim that the method broadly solves the long-tail issue.


**Presentation:** The paper is easy to follow. The motivation, pipeline, and main results are presented in a straightforward way, and Figure 1 is helpful. The table showing head versus long-tail gaps is good.

**Suggestions:**
1. The exact inference pipeline and what model finally produces the scene graph **after the ICL prompt are not always explained** as explicitly as they should be.
2. A few implementation choices feel underexplained, such as why top-1 is best, how the router prompts are finalized, and how sensitive results are to threshold choices beyond the shown plot. (Is it dataset specific hyperparameter)?

The writing is readable in a structured manner, but more detail would be needed for full reproducibility of the work.


**Significance:** The authors addresses an important problem for surgical multimodal scene understanding in operating room setting. The measured improvement is still moderate overall, and fairly specialized in a specific data domain. So, I see the significance as meaningful but somewhat bounded: useful for this subarea, but not yet a broadly transformative result.

**Originality:** The work is moderately original. The individual ingredients are not entirely new: routing, retrieval, reranking (BERT), and ICL (Brown et. Al, 2020: Language Models are Few-Shot Learners) are all familiar ideas. The novelty of the present work primarily lies in how these components are combined to address long-tail relation prediction for multimodal OR scene graph generation. Introducing a scene-graph-aware reranking target based on weighted triplet IoU, on long-tail relations in this domain is a reasonable contribution. Though it feels more like a careful system design for a new setting than a fundamentally new algorithmic idea.

---

> ### Author Rebuttal · Authors · 2026-03-29
>
> 1. Thank you for this valuable suggestion. Following your suggestion, we repeated the MM-OR experiment with five different random seeds. We report the experimental results below in the format of $\mathit{mean}_{\mathit{standard\ deviation}}$.
> | Method | Head Classes |  |  | Long-Tail Classes |  |  | Macro Average |  |  |
> |:---:|:---:|:---:|:---:|:---:|:---:|:---:|:---:|:---:|:---:|
> |  | Precision | Recall | F1 | Precision | Recall | F1 | Precision | Recall | F1 |
> | MM2SG | **0.759** |**0.663**| **0.699** | 0.449 | 0.268 | 0.289 | 0.623 | 0.490 | 0.520 |
> | SGG-ICL  | $0.752_{0.001}$| $0.655_{0.002}$ | $0.692_{0.002}$ | **$0.528_{0.003}$** | **$0.334_{0.003}$** | **$0.357_{0.003}$** | **$0.653_{0.003}$** | **$0.515_{0.001}$** | **$0.544_{0.003}$** |
>
>
> 2. We added two simpler retrieval baselines. Specifically, ‘Random ICL’ refers to using randomly selected demonstrations, while ‘w/o MLLM Reranker’ refers to using BGE-VL-MLLM for multimodal retrieval and selecting the top-1 result as the ICL demonstration. The experimental results are as follows:
> | Method | Head Classes |  |  | Long-Tail Classes |  |  | Macro Average |  |  |
> |:---:|:---:|:---:|:---:|:---:|:---:|:---:|:---:|:---:|:---:|
> |  | Precision | Recall | F1 | Precision | Recall | F1 | Precision | Recall | F1 |
> | Random ICL | 0.745 | 0.642 | 0.685 | 0.322 | 0.302 | 0.271 | 0.559 | 0.493 | 0.503 |
> | w/o MLLM Reranker | 0.752 | 0.654 | 0.690 | 0.368 | 0.331 | 0.298 | 0.584 | 0.513 | 0.519 |
> | SGG-ICL | **0.753** | **0.657** | **0.693** | **0.530** | **0.336** | **0.358** | **0.655** | **0.516** | **0.546** |
>
> 3. The inference time of MM2SG is approximately 1 second per sample. For head-class samples, our method takes approximately 3 seconds per sample, since they only go through the routing stage and no ICL inference is performed. For long-tail-class samples, which account for only about one-tenth of the dataset, our method takes approximately 5 seconds per sample.
>
> 4. For some long-tail classes, such as Cutting, there are only four samples in the test set. Our adaptive router successfully routed two of them as long-tail classes and retrieved the correct ICL demonstrations for them. However, the subsequent ICL inference still failed to predict the Cutting relation contained in those samples. We think the failure is due to the fact that, for certain extremely scarce long-tail classes, the final SGG model is still unable to make correct inferences even with ICL.

---

> > ### Author Rebuttal · Reviewer_BZEN · 2026-04-02
> >
> > My concerns have been adequately addressed by the author's responses. I hereby retain my score.

---

> > > ### Author Response · Authors · 2026-04-02
> > >
> > > We are very pleased that our rebuttal has addressed your concerns. Since the issues you raised have now been fully resolved, we would sincerely appreciate your consideration of a higher score.

---

### Official Review · Reviewer_m8g1 · 2026-03-13

**Soundness:** 2
**Presentation:** 2
**Significance:** 2
**Originality:** 2
**Overall Recommendation:** 4
**Confidence:** 4

**Summary:**

This paper introduces SGG-ICL, a novel framework designed to tackle the severe long-tail distribution problem in operating room scene graph generation. Recognizing that existing Multimodal Large Language Models struggle to recognize infrequent surgical actions, the authors propose a selective In-Context Learning pipeline. The framework operates in three main steps: First, an Adaptive Router uses a large vision-language model  to classify whether a given query involves a long-tail category. Second, for identified long-tail queries, a coarse pre-selection retrieves structurally similar training examples using text and images. Finally, a custom MLLM Reranker trained using an iou similarity metric derived from ground-truth triplets evaluates the full multimodal inputs  to select the optimal top-$k$ examples. These examples are then used as in-context prompts to aid the final scene graph generation.

**Compliance With Llm Reviewing Policy:**

Affirmed.

**Key Questions For Authors:**

Please see the weaknesses above.

**Limitations:**

The paper would benefit from a clearer discussion of limitations, particularly regarding the computational requirements of the multi-model pipeline and the risks of errors when deploying automated scene understanding systems in surgical settings.

**Strengths And Weaknesses:**

pos:

- The paper introduces a tailored in-context learning framework for long-tail scene graph generation in surgical operating rooms, including a novel MLLM reranker trained with weighted triplet IoU to measure structural graph similarity.
- The framework is clearly structured, and the Adaptive Router is a practical component that selectively applies ICL to long-tail samples to avoid harming head-class performance and reduce context overhead.

cons:

- The pipeline relies on several large models (such as Qwen3-VL-30B and multiple MLLMs), raising concerns about latency, memory cost, and real-time deployment feasibility in operating room scenarios.
- The paper does not clearly explain how the MLLM reranker evaluates structural similarity at inference when the test query has no ground-truth graph, making the prompting mechanism unclear.
- Using a 30B-parameter model for a binary routing decision appears computationally excessive, and the paper lacks ablations exploring lighter alternatives.
- The method appears closely tied to the MM-OR dataset and predefined action categories, and its transferability to other surgical datasets or unseen categories is not demonstrated.

---

> ### Author Rebuttal · Authors · 2026-03-29
>
> 1. The inference time of MM2SG is approximately 1 second per sample. For head-class samples, our method takes approximately 3 seconds per sample, since they only go through the routing stage and no ICL inference is performed. For long-tail-class samples, which account for only about one-tenth of the dataset, our method takes approximately 5 seconds per sample.
>
> 2. During inference, the MLLM reranker does not require the ground-truth graph of the test query. The input to the MLLM reranker is the multimodal information from two scenes, and it outputs a similarity score between the scene graphs of these two scenes as inferred by the model.
>
> 3. Our Adaptive Router uses Qwen3-VL-30B, while the MLLM Reranker uses LLaVA-7B. The Adaptive Router is kept frozen, and only the MLLM Reranker is trained. We replaced the large model in the Adaptive Router with Qwen3-VL-8B-Instruct, and the experimental results on the MM-OR dataset are as follows:
> | Method | Head Classes |  |  | Long-Tail Classes |  |  | Macro Average |  |  |
> |:---:|:---:|:---:|:---:|:---:|:---:|:---:|:---:|:---:|:---:|
> |  | Precision | Recall | F1 | Precision | Recall | F1 | Precision | Recall | F1 |
> | MM2SG | **0.759** |**0.663**| **0.699** | 0.449 | 0.268 | 0.289 | 0.623 | 0.490 | 0.520 |
> | SGG-ICL (8B) | 0.748 | 0.645 | 0.686 | 0.514 | 0.318 | 0.341 | 0.645 | 0.501 | 0.535 |
> | SGG-ICL (30B) | 0.753 | 0.657 | 0.693 | **0.530** | **0.336** | **0.358** | **0.655** | **0.516** | **0.546** |
>
> The experimental results show that the 8B version still achieves a 5.2% improvement in F1 score on long-tail classes.
>
> 4. Since MM-OR is currently the only publicly available dataset for OR SGG, we conducted experiments only on the MM-OR dataset. Moreover, like our method, other OR SGG works, such as Oracle and MM2SG, are also evaluated only on OR SGG datasets. MM-OR is a large-scale OR scene graph generation dataset containing 92,983 total time points and 25,277 annotations, amounting to 500 GB of data.

---

> > ### Author Rebuttal · Reviewer_m8g1 · 2026-04-08
> >
> > Thanks for the author's rebuttal. My concerns have been adequately addressed and I will keep my positive score.

---

> > > ### Author Response · Authors · 2026-04-08
> > >
> > > We are very pleased that our rebuttal has addressed your concerns. Since the issues you raised have now been fully resolved, we would sincerely appreciate your consideration of a higher score.

---

### Decision · Program_Chairs · 2026-04-30

**Decision:**

Accept (regular)

**Comment:**

This submission addresses long-tail operating room scene graph generation and proposes a selective in-context learning pipeline centered on an adaptive router and a multimodal reranker. Reviewers generally agreed that the problem is meaningful and of clear practical relevance, and several found the route–retrieve–rerank framework coherent, well motivated, and appropriately designed for the task. The paper was particularly credited for improving long-tail performance while largely preserving head-class behavior, and for presenting a technically thoughtful system tailored to a challenging real-world application.

The main concerns centered more on scope and deployability than on any fundamental weakness in the core idea. In particular, reviewers raised questions about the reliance on multiple large models, the lack of a more explicit latency and resource analysis for operating-room deployment, evaluation on only a single dataset, and limited evidence regarding robustness, run-to-run variance, and failure modes. There was also some disagreement over the extent to which the contribution goes beyond a carefully engineered domain-specific system.

The rebuttal clarified several implementation details and provided additional empirical support. Some reviewers explicitly indicated that their concerns had been sufficiently addressed, while others retained partial reservations. Overall, the paper makes a meaningful contribution to a practically important problem, and the remaining concerns appear to reflect limitations in scope and efficiency rather than decisive issues of validity or technical soundness. I therefore recommend acceptance.